# Nondestructive Monitoring Hydration of Belite Calcium Sulfoaluminate Cement by EIS Measurement

**DOI:** 10.3390/ma15134433

**Published:** 2022-06-23

**Authors:** Lin Chi, Mian Wang, Zhuolin Wang, Zhenming Li, Bin Peng, Junjie Li

**Affiliations:** 1School of Environment and Architecture, University of Shanghai for Science and Technology, Shanghai 200093, China; chilin@usst.edu.cn (L.C.); 213382216@st.usst.edu.cn (M.W.); 1935023313@st.usst.edu.cn (J.L.); 2Shanghai Key Laboratory of Engineering Structure Safety, Shanghai Research Institute of Building Sciences Co., Ltd., Shanghai 200032, China; wzllzw@163.com; 3Microlab, Faculty of Civil Engineering and Geoscience, Delft University of Technology, 2628 CN Delft, The Netherlands

**Keywords:** belite calcium sulfoaluminate cement, resistivity, cement hydration, EIS measurement

## Abstract

In this study, the impact of water-to-cement (*w*/*c*) ratios of belite calcium sulfoaluminate cement (BCSA) on the hydration kinetics and the electrochemical impedance spectroscopy (EIS) parameters is studied. According to the analysis of classic hydration measurements, such as calorimetry tests, chemical shrinkage content, and chemically bound water content, it can be concluded that a higher *w*/*c* ratio clearly accelerates the hydration of BCSA cement paste. The electrical resistivity of BCSA0.35 cement paste is more than 4.5 times that of BCSA0.45 and BCSA0.5, due to the gradually densified micropore structure blocking the electrical signal transmission rather than the free charged-ion content. The porosity of BCSA0.5 is 27.5% higher than that of BCSA0.35 and 7.8% higher than that of BCSA0.45, which proves the resistivity is clearly related to the variation in microstructure, especially for the porosity and pore size distribution. The novelty of this study is the linear regression with logarithm terms of electrical resistivity and classic hydration parameters such as chemical shrinkage, cumulative hydration heat, and chemically bound water is established to extend the classical expression of cement hydration degree. It indicates that the electrochemical impedance spectroscopy can be taken as a nondestructive testing measurement to real-time monitor the cement hydration process of cement-based materials.

## 1. Introduction

As the primary binder material in the construction industry, cement is produced and consumed dramatically, especially in developing countries [1,2]. The cement industry is upgrading to produce new types of cement to reduce CO_2_ emissions and energy consumption to improve resource utilization [3,4,5,6]. Belite calcium sulfoaluminate (BCSA) cement is composed of belite, calcium sulfoaluminate, ferrite, and calcium sulfate [3,7]. As the main phase in BCSA cement, C2S has a lower firing temperature in the kiln (typically around 1250 °C), which decreases CO_2_ emissions by 12–15% compared with ordinary Portland cement [8,9].

Compared with OPC cement, the hydration process of the BCSA cement system is slightly different. As the basic mineral composition, calcium sulfoaluminate reacted with calcium sulfate and forms ettringite and aluminum hydroxide, which provides the early age mechanical strength for the cement matrix. When calcium sulfoaluminate is almost consumed, the monosulphoaluminate is formed. C-S-H gel and portlandite are the major hydration products of C2S, which provide long-age mechanical strength and better durability for cementitious materials [10,11,12].

In the last few years, nondestructive testing measurements (NDT), such as dynamic or vibration tests, acoustic emission techniques, infrared image methods, and ultrasonic methods have been applied to assess the damage and infernal defects of civil engineering structures [13,14,15,16,17,18,19,20]. The optimization of concrete detection technology effectively improves the accuracy of detection and reduces the quality of concrete problems aroused by the building safety hazards, promoting the improvement of the quality of construction projects. As one of the NDT methods, the electrochemical impedance spectroscopy (EIS) test is considered an effective test method to analyze the early performance of cement-based materials [21,22,23,24,25].

Microstructure analysis, such as X-ray diffraction (XRD), scanning electron microscopy (SEM), hydration heat analysis, and thermogravimetric analysis (TGA), is commonly used to study the early hydration characteristics of cement-based materials [26,27]. However, except for hydration heat analysis, these testing methods can only capture the hydration characteristics of cement-based materials at a certain moment, which cannot monitor the early hydration process of cement-based materials in real-time without damage, and it is difficult to accurately predict and evaluate the mechanical strength and durability of cement-based materials at a later stage. Based on the EIS method, real-time monitoring of early hydration performance of cement-based materials and the microstructure variation prediction can be solved. However, few studies focus on the hydration assessment by the EIS measurement of cement-based materials, especially for the BCSA cement system [28].

According to our previous studies [29,30], the microstructure development of the BCSA cement-based material system can be characterized by the EIS method. A modified linear equation involving resistivity, chemical shrinkage, chemically bonded water, and hydration heat release is proposed. However, that study was only focused on cement paste with various additives, with the influence of *w*/*c* ratios unclarified. The aim of this work is to clarify the impact of the *w*/*c* ratio on the hydration kinetics and the electrochemical performance of the BCSA cement paste. The novelty of this study is the potential correlations between traditional hydration parameters, such as chemical shrinkage, chemically bound water, hydration heat, and electrical resistivity.

## 2. Materials and Methods

### 2.1. Materials

BCSA cement was produced by Tianjin Cement Co., Ltd. The chemical compositions of BCSA cement obtained by X-ray fluorescence spectrometry (XRF, PW4400, Panalytical Inc.) are shown in Table 1. The Blaine fineness and absolute density of BCSA cement is 3700 cm^2^/kg and 3.10 g/cm^3^, respectively. Three *w*/*c* ratios (0.34, 0.45 and 0.5) are applied for BCSA cement paste.

### 2.2. Hydration Heat

The hydration heat of BCSA cement pastes was detected by a TAMAIR isothermal calorimeter at the temperature of 21 °C. With the distilled water injected into the vessel, the paste was inner-mixed for 180 s in the calorimetry bottle. Isothermal calorimetry test was monitoring the kinetics of cement hydration in the first 72 h.

### 2.3. Chemical Shrinkage

According to ASTM C 1608-12, the chemical shrinkage content of BCSA cement paste was measured [31]. In total, 10 g cement paste was filled in a 20 ml vial (*ϕ* 27.5 mm, H 57 mm) and vibrated for 30 s on a vibrating table. Distilled water was filled in the remaining space of the vial. A rubber stopper with a 1 ml capillary tube embedded was set at top of the vial. The paraffin oil was instilled in the capillary tube in case of water evaporation. The whole set was transferred into a water bath at 21 ± 1 °C. The waterline of the capillary tubes was recorded at 15 min as the starting line and once a day for 28 d. For each ratio, the chemical shrinkage of three samples was measured for average.

### 2.4. Chemically Bound Water

Freshly mixed cement paste was cast in 2 cm × 2 cm × 2 cm mold, demolded after 24 h and cured in the climate chamber with 65% RH, 21 °C until further testing. The degree of hydration of each binder was tested on days 1, 3, 7, and 28, respectively.

Cement paste was smashed to 3 mm grains and stopped hydration by the acetone-methanol mixtures. In the BCSA cement system, samples were dried in a vacuum drying chamber at 50 °C instead of 105 °C in the case of ettringite transformation. Then, the samples were calcined at 1050 °C in the furnace for 1 h. According to Powers’ model [32], chemically bound water at a certain age can be calculated by the following equation:(1)$$m_{\mathrm{chem}}=\frac{m_{50}-m_{1050}}{m_{50}}\times 100\%$$
where *m*_50_ or *m*_1050_ is the cement mass after drying at 105 °C or 1050 °C.

### 2.5. Compressive Strength

Samples with the dimension of 20 mm× 20 mm × 20 mm were cast and demolded after 24 h, then cured in the curing room with 21 ± 1 °C and RH > 65%. The compressive strength of BCSA cement paste was measured with the loading rate of 2400 N/s ± 200 N/s on days 3, 7, and 28 according to ASTM C 109/109M-21 [33].

### 2.6. Pore Size Distribution

The pore structure of BCSA cement samples was measured on day 28 by Mercury intrusion porosimetry (MIP). MIP was measured by Micromeritics AutoPore IV9500 (Micromeritics Instrument Corporation) with a contact angle of 140° at 25 °C. The maximum and minimum pressures were set at 414 MPa and 1.4 kPa, respectively. The samples were cut into 5 mm cubes and stopped the cement hydration by the acetone-solvent exchange for 12 h.

### 2.7. EIS Measurement

Fresh mixed BCSA cement pastes were cast in the acrylic molds (65 mm × 30 mm × 47 mm) [34]. Two stainless steel-welded meshes (47 mm × 47 mm) with a distance of 40 mm were served as electrodes and embedded into BCSA cement paste. The experimental setup of EIS measurement of the BCSA cement paste is present in Figure 1. The EIS experiment was measured by an electrochemical workstation (VersaSTAT 4000 A Princeton). The frequency is set at 10^5^–0.1 Hz and the amplitude is set at 10–100 mV [35]. The measurement was taken during different hydration periods (1–12 h, 1–28 d). ZsimpWin 3.5 (Echem Software) is integrated with the VersaStudio software (Princeton Applied Research) to analyze Nyquist and Bode diagrams.

## 3. Results

### 3.1. Hydration Heat Release

Figure 2 shows the hydration heat release of BCSA cement paste with various *w*/*c* ratios. The initial heat release is corresponding to the dissolution heat. The induction stage at around 4.5 h is considered a nucleation and growth process. A sharp exothermic peak was followed and characterized to be the acceleration of ettringite formation. The acceleration stage continues until the sulfate ions in the solution were consumed [30,36]. It is observed that the main exothermic peak became stronger in BCSA0.5 (in Figure 2a). For the cumulative heat release (Figure 2b), the effect of *w*/*c* showed similar trends on BCSA cement pastes. The hydration rate of BCSA0.35 is relatively low. Certain water content is needed for the formation of hydration products. With a relatively higher amount of water, the less the nuclear site is and hydrates cannot grow larger to cover the spatial gap between each particle and form the microstructure of the cement matrix.

### 3.2. Chemically Bound Water

Isothermal calorimetry measurement is mainly applied to investigate the early hydration kinetics of BCSA cement paste and chemically bound is applied to evaluate the hydration kinetics for the long age [30]. The chemically bound water contents of BCSA cement paste at days 1–28 are shown in Figure 3. The chemically bound water content of all BCSA pastes distinctly increases at days 1–3 and then stabilizes at days 3–28. It is due to a relatively higher hydration rate of the ye’elimite phase at an early age and a low hydration rate of C_2_S at days 3–28. The chemically bound water of BCSA0.5 is relatively higher than BCSA0.35 or 0.45, which is consistent with the calorimetry results.

### 3.3. Chemical Shrinkage

The chemical shrinkage content of BCSA cement pastes with different *w*/*c* ratios at the age of 1 d–28 d is shown in Figure 4. There is little difference in chemical shrinkage behavior of BCSA pastes with different *w*/*c* ratios at 0–5 h. The chemical shrinkage content of BCSA0.5 is slightly higher than that of BCSA0.35 and BCSA0.45 at days 1–10. After day 10, the chemical shrinkage content of all binders tends to be stable at around 0.10 mL/g. For an open water system, the chemical shrinkage content of BCSA pastes of different *w*/*c* ratios is hard to measure. Considering the applicability of the three test methods discussed above, the EIS technique is applied to monitor and characterize the whole hydration stage of cement [37,38,39].

### 3.4. Compressive Strength

Figure 5 presents the compressive strength of BCSA paste with different *w*/*c* ratios. It is obvious that the compressive strength of BCSA cement paste increases with the curing age and decreases with the *w*/*c* ratios. The compressive strength of BCSA0.35 at 3 d is increased by 158.4% than that of BCSA0.45 and 167.0% than that of BCA0.5. The compressive strength of BCSA0.35 at 28 d is increased by 18.6% than that of BCSA0.45 and 132.5% than that of BCA0.5. In general, the *w*/*c* ratio has a strong effect on the compressive strength of BCSA cement paste, especially at the early age of 3 d.

### 3.5. EIS Measurement

Nyquist and Bode curves and fitting curves of BCSA0.35, 0.45 and 0.50 at 1 d, 3 d, 7 d, 14 d, and 28 d are present in Figure 6. The radius of the high-frequency arc increases with hydration age [29,40]. The equivalent circuit model *R*_*s+l*_ (*R*_*int*_*C*_*int*_) (*R*_ct_*C*_dl_) is applied to analyze the impedance spectrum [41]. According to the above equivalent circuit, *R_s+l_* and *R_int_* can be extracted [15,42,43]. Compared with *R_int_*, *C_int_* is several orders of magnitude lower and is not counted. Therefore, as the total value of *R_s+l_* and *R_int_*, *R*_s_ is taken as an important parameter to evaluate the hydration reaction and transport properties of the BCSA cement paste [17,44]. Electrochemical parameters of equivalent circuit R {CR} {CR} in Nyquist curves Figure 6 are presented in Appendix A.

The electrical resistivity of BCSA cement pastes with various *w*/*c* ratios at days 1–14 and day 28 is shown in Figure 7. The resistivity of BCSA cement paste increases corresponding to the curing age. The electrical resistivity curve of all samples is basically the same in the first 24 h and the charge transfer only depends on the electrons in the pore solution [45]. After 24 h, the electrical resistivity of all samples decreases with increasing *w*/*c* ratio. With the microstructural growth and the accumulation of hydration products, the porosity and pore connectivity become lower [46]. Additionally, the gradually densified the micropore structure block electrical signal transmission [16]. These variations in the microstructure of cement-based materials can be expressed in terms of resistivity; therefore, the EIS measurement is a simple but efficient method to characterize the microstructure variation [47].

### 3.6. Porosity Measurement

The pore structure and pore size distribution of cement paste BCSA0.35, BCSA0.45 and BCSA0.50 are measured by MIP measurement are shown in Table 2 and Figure 8. The porosity of BCSA0.5 is 27.5% higher than that of BCSA0.35 and 7.8% higher than that of BCSA0.45.

Since the conductivity of solid and vapor phases are several orders of magnitude lower than the liquid phase [48,49,50], it is assumed that the electrical conduction in the cement matrix only occurs through its liquid phases. Therefore, ion concentration in the liquid phase and tortuosity of cement microstructure are two key factors affecting the electrical resistivity of the cement system, which can be expressed by Equation (2) [51].
(2)$$\rho_{t}=\beta^{-1}\phi^{-1}\rho_{0}$$
where *ρ*_0_ is the resistivity of the liquid phase, *ρ_t_* is the electrical resistivity of the bulk cement paste, *ϕ* is the total porosity of liquid-filled pores, and *β* is the pore connectivity and is equal to the inverse of tortuosity.

With hydration products constantly fill the microstructure in the cement matrix, a continuous decrease in the total porosity is observed. The variation in the pore structure of BCSA cement paste is consistent with that of resistivity, which proves that resistivity can reflect the porosity, the average pore size, and the ion concentration of the pore solution during the cement hydration process.

## 4. Discussions

### 4.1. Correlations between Electrical Resistivity and Hydration Heat Release

Figure 9 presents the relation curve of the hydration heat release and electrical resistivity of BCSA cement pastes with various *w*/*c* ratios. The linear fitting curve and the correlation equation of the total heat release from hydration heat and the resistivity are established in Equation (3). The correlation coefficient is close to 1. The relationship between volume resistivity and cumulative heat release of the BCSA cement paste can be induction as Equation (4). This result indicates that resistivity extracted from the EIS test can be regarded as a real-time monitoring parameter and applied to measure the hydration heat of mass concrete structures in the future.
*Q* (t) = 71.43 ln*ρ*_0.35_(t) − 427.64 *R*^2^ = 0.988
*Q* (t) = 90.91 ln*ρ*_0.45_(t) − 547.45 *R*^2^ = 0.996
*Q* (t) = 113.63 ln*ρ*_0.50_(t) − 694.20 *R*^2^ = 0.996(3)
*Q* (t) = *f*_1_(*r*) ln*ρ_r_* (t) + *f*_2_(*r*)
*f*_1_(*r*) = 268.97 *r* − 24.56 *R*^2^ = 0.946
*f*_2_(*r*) = 1694.4 *r* − 177.79 *R*^2^ = 0.940(4)
where *ρ_r_
*(t) is the electrical resistivity at a certain time, in Ω cm; *Q* (t) is the cumulative hydration heat at time t, in J/g; *r* is the *w*/*c* ratio; and *f*_1_(*r*) and *f*_2_(*r*) are the slope and intercept, respectively, and are the functions of the *w*/*c* ratio.

### 4.2. Correlations between Electrical Resistivity and Chemical Shrinkage

Linear correlations between chemical shrinkage and the logarithm of the electrical resistivity of BCSA pastes with various *w*/*c* ratios are present, see Figure 10. By further induction, the relationship equation between bulk electrical resistivity and chemical shrinkage of hardened BCSA cement pastes can be obtained, as shown in Equation (6). With the increase in the *w*/*c* ratio, the slope and intercept of the characteristic function increase correspondingly and have a good linear relationship with the *w*/*c* ratio. Therefore, the corresponding chemical shrinkage content of cement-based materials can be obtained by monitoring the resistivity variation.
*Cs* (t) = 0.02 ln*ρ*_0.35_(t) − 0.12 *R*^2^ = 0.999
*Cs* (t) = 0.03 ln*ρ*_0.45_(t) − 0.18 *R*^2^ = 0.997
*Cs* (t) = 0.04 ln*ρ*_0.5_ (t) − 0.22 *R*^2^ = 0.992(5)
*Cs* (t) = *f*_3_(*r*) ln*ρ_r_* (t) + *f*_4_(*r*) (6)
*f*_3_(*r*) = 0.129 *r* − 0.026 *R*^2^ = 0.964
*f*_4_(*r*) = 0.657 *r* − 0.111 *R*^2^ = 0.994(6)
where *Cs* (t) is the chemical shrinkage content of BCSA cement paste at a certain time, in mL/g; and *f*_3_(*r*) and *f*_4_(*r*) are the slope and intercept, respectively, and are the functions of the *w*/*c* ratio.

### 4.3. Correlations between Electrical Resistivity and Chemically Bound Water

The linear relationship between the logarithm of resistivity and the chemically bound water of the BCSA cement pastes with different *w*/*c* ratios is shown in Figure 11. By further induction, the relationship equation between bulk electrical resistivity and chemically bound water of hardened BCSA cement pastes can be obtained, as shown in Equation (8).
*Cw* (t) = 0.02 ln*ρ*_0.35_(t) − 0.12 *R*^2^ = 0.980
*Cw* (t) = 0.04 ln*ρ*_0.45_(t) − 0.21 *R*^2^ = 0.949
*Cw* (t) = 0.06 ln*ρ*_0.5_ (t) − 0.42 *R*^2^ = 0.959(7)
*Cw* (t) = *f*_5_(*r*) ln*ρ_r_* (t) + *f*_6_(*r*)
*f*_5_(*r*) = 0.257 *r* − 0.071 *R*^2^ = 0.964
*f*_6_(*r*) = 1.843 *r* − 0.549 *R*^2^ = 0.840(8)
where *Cw* (t) is the chemically bound water content of cement paste at a certain time, in mL/g; *m*_1_ and *m*_2_ are the fitting parameters; and *f*_5_(*r*) and *f*_6_(*r*) are the slope and intercept, respectively, and are the functions of the *w*/*c* ratio.

### 4.4. Hydration Degree Evaluation

The chemical shrinkage content, cumulative heat release content, and chemically bound water content are all important parameters that characterize the degree of cement hydration. Additionally, the relationship between Ln*ρ*(t) and *Q*(*t*), *Cs*(*t*), or *CW*(*t*) (Equations (4), (6), and (8)) belongs to linear regression with logarithmic terms and can be deduced as Equation (9).
(9)$$\alpha =\frac{Q_{n}\left(t\right)}{Q_{n}^{0}}=\frac{CW\left(t\right)}{CW_{}^{0}}=\frac{CS\left(t\right)}{CS_{}^{0}}=\frac{Ln\rho \left(t\right)}{Ln\rho_{}^{0}}$$
where *Q*(*t*) or $Q_{n}^{0}$ is the cumulative hydration heat at time t or full hydration, in J/g; *CW*(*t*) or $CW_{}^{0}$ is the chemically bound water content of cement paste at a certain time or full hydration, in mL/g; *CS*(*t*) or $CS_{}^{0}$ is the chemical shrinkage content of cement paste at a certain time or full hydration, in mL/g; and *ρ*(*t*) or $Ln\rho_{}^{0}$ is the electrical resistivity at a certain time or full hydration, in Ω cm.

Based on the variation in electrical resistivity or electrical impedance, electrochemical impedance spectroscopy measurement can be applied to characterize the microstructure evolution of cement-based materials in the early hydration stage. Additionally, the real-time monitoring properties of cement-based materials can be obtained by establishing the correlation between resistivity and major physical and chemical performance parameters. Additionally, EIS measurement can enhance the accuracy of the hydration prediction model of belite calcium sulfoaluminate cement, especially for the early hydration age.

## 5. Conclusions

Electrical impedance spectroscopy was applied to monitor the hydration of belite calcium sofoaluminate cement with a *w*/*c* ratio of 0.35, 0.45, and 0.50. Additionally, the main conclusions are summarized as follows.

(1)Higher *w*/*c* ratios clearly accelerate the hydration of BCSA cement paste. The porosity of BCSA0.5 is 27.5% higher than that of BCSA0.35 and 7.8% higher than that of BCSA0.45.(2)The electrical resistivity of BCSA0.35 cement paste is more than 4.5 times that of BCSA0.45 and BCSA0.5, due to gradually densified the micropore structure blocking the electrical signal transmission rather than the free charged-ion content. The resistivity is clearly related to the variation in the microstructure, especially for the porosity and pore size distribution during the hydration process.(3)Linear regression with logarithmic terms of electrical resistivity and classic hydration parameters, such as chemical shrinkage, cumulative hydration heat, and chemically bound water, is built to express the cement hydration degree.

## Figures and Tables

**Figure 1 materials-15-04433-f001:**
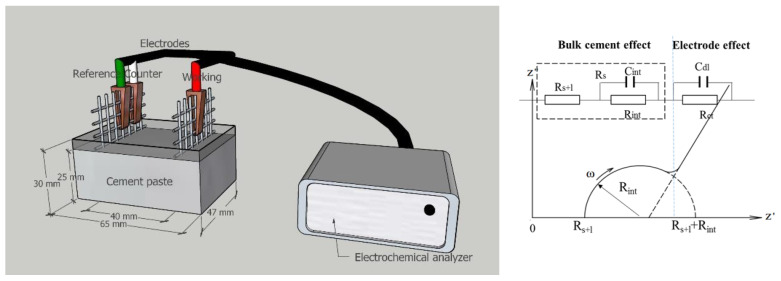
Experimental setup of EIS measurement of the BCSA cement paste and the equivalent circuit model [29].

**Figure 2 materials-15-04433-f002:**
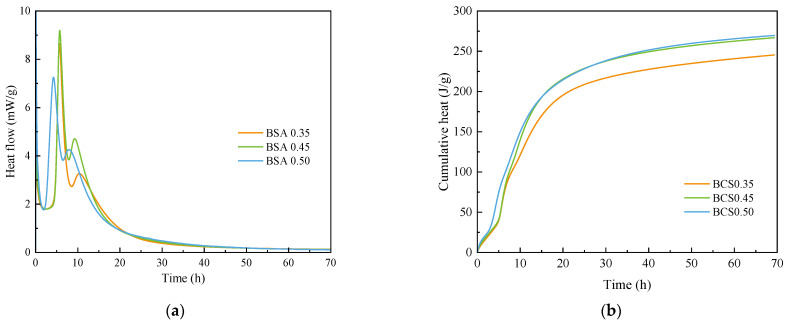
Heat flow (**a**) and cumulative heat curves (**b**) of BCSA cement pastes within 72 h.

**Figure 3 materials-15-04433-f003:**
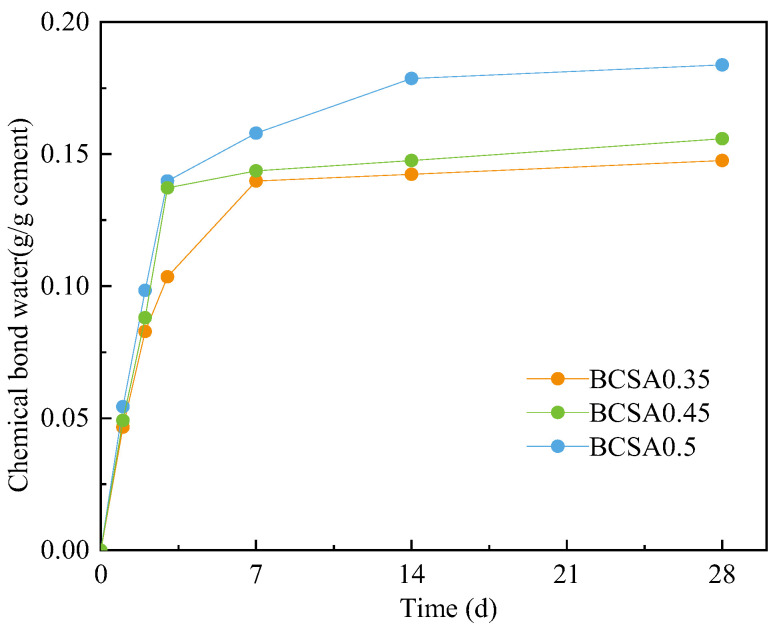
Chemically bound water content of BCSA cement pastes at days 1–28.

**Figure 4 materials-15-04433-f004:**
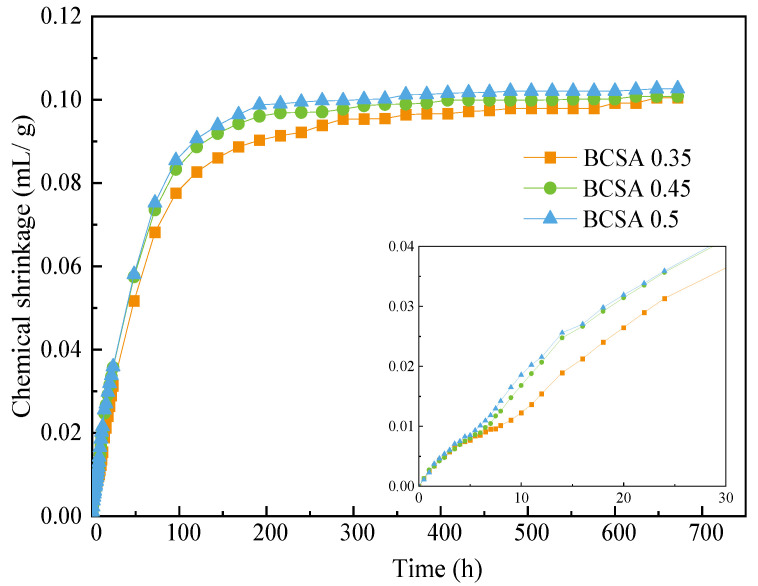
Chemical shrinkage of BCSA pastes with different *w*/*c* ratios.

**Figure 5 materials-15-04433-f005:**
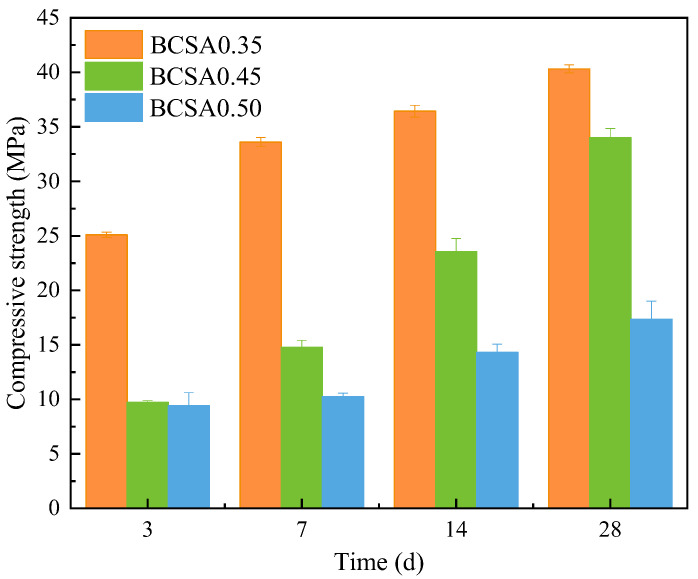
Compressive strength of BCSA pastes with different *w*/*c* ratios.

**Figure 6 materials-15-04433-f006:**
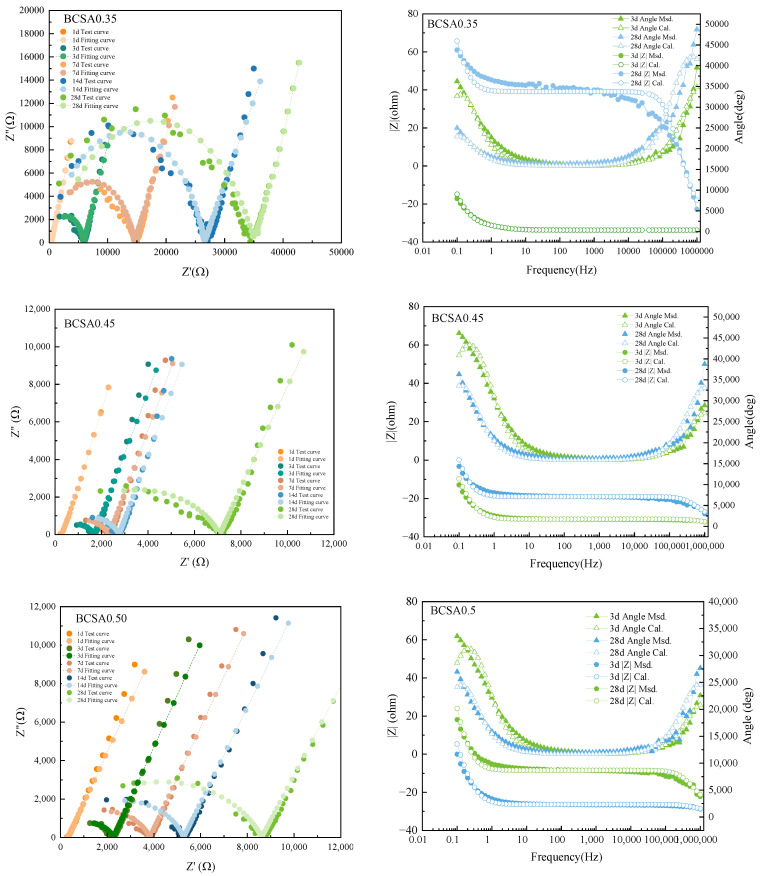
Nyquist and Bode curves and fitting curves of BCSA0.35, 0.45 and 0.50 at 1 d, 3 d, 7 d, 14 d, and 28 d.

**Figure 7 materials-15-04433-f007:**
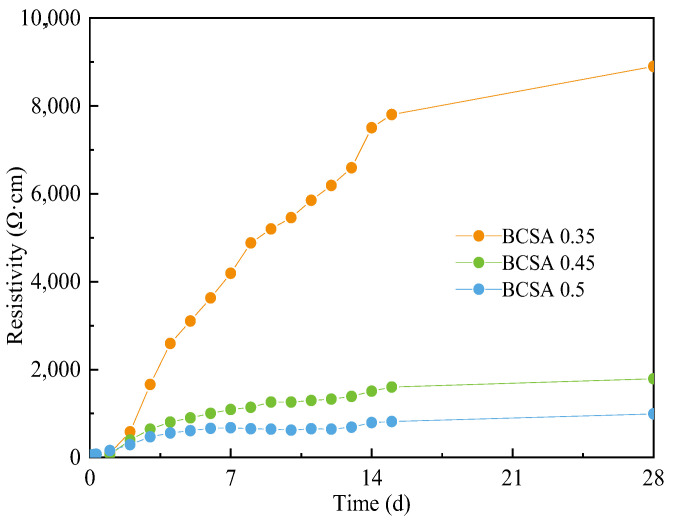
The electrical resistivity of BCSA cement pastes with various *w*/*c* ratio.

**Figure 8 materials-15-04433-f008:**
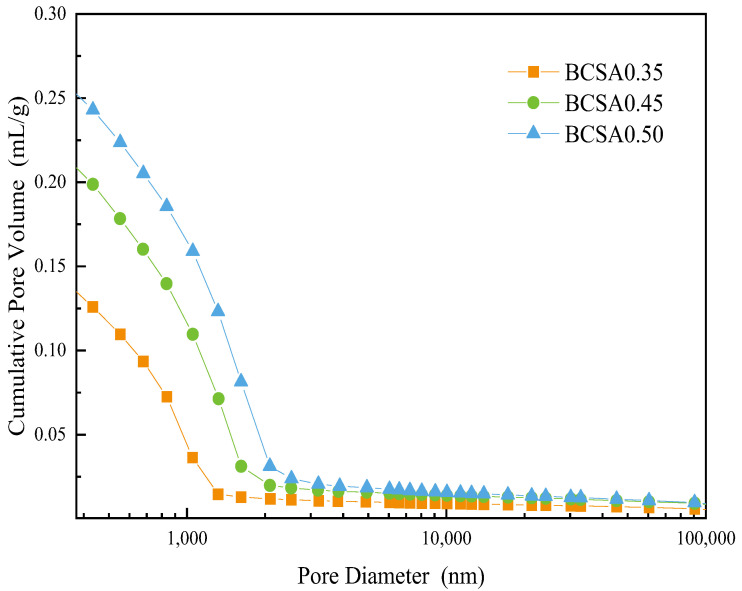
Pore size distribution of BCSA cement pastes with various *w*/*c* ratios at 28 d.

**Figure 9 materials-15-04433-f009:**
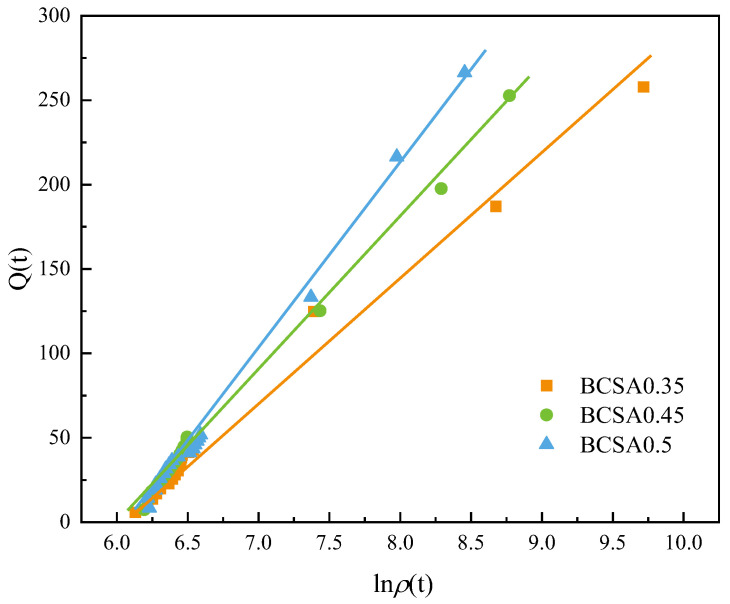
Correlation between cumulative hydration heat and electrical resistivity of BCSA pastes with various *w*/*c* ratios.

**Figure 10 materials-15-04433-f010:**
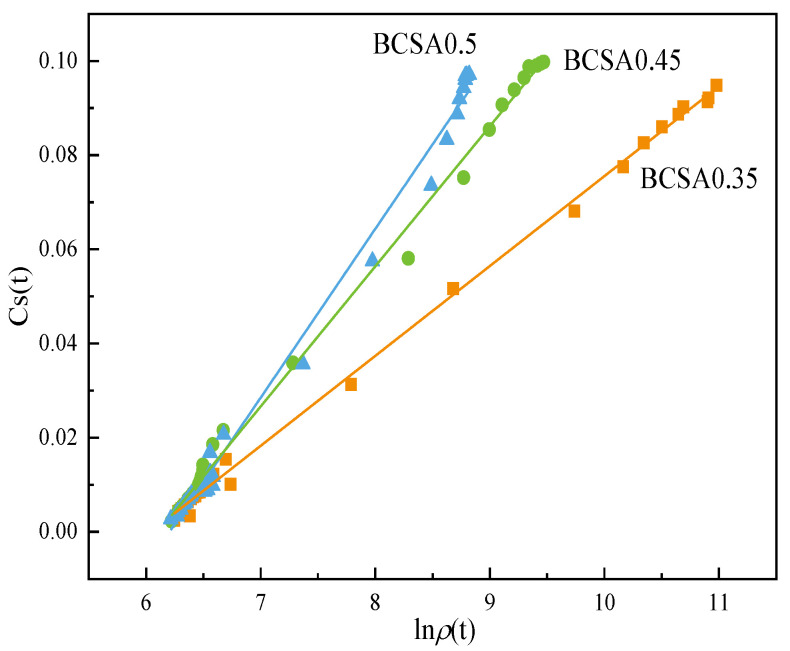
Correlation between chemical shrinkage content and electrical resistivity of BCSA pastes with various *w*/*c* ratios.

**Figure 11 materials-15-04433-f011:**
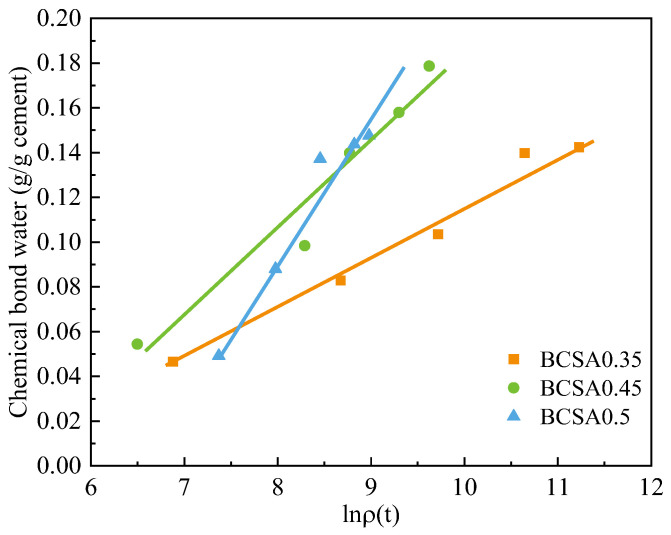
Correlation between chemically bound water and electrical resistivity of BCSA pastes with various *w*/*c* ratios.

**Table 1 materials-15-04433-t001:** Chemical composition of BCSA cement (wt.%).

SiO_2_	CaO	Fe_2_O_3_	Al_2_O_3_	Na_2_O	K_2_O	MgO	SO_3_	LOI
14.16	49.6	2.69	21.65	0.13	0.25	0.96	9.54	1.02

**Table 2 materials-15-04433-t002:** Porosity and tortuosity of BCSA cement pastes.

	BCSA0.35	BCSA0.45	BCSA0.50
Porosity	38.01	44.94	48.45
Tortuosity	88.61	63.82	51.59

## Data Availability

Not applicable.

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
