# Peer review of "Nondestructive Monitoring Hydration of Belite Calcium Sulfoaluminate Cement by EIS Measurement"

_materials, 2022, doi:10.3390/ma15134433_

Round 1

Reviewer 1 Report

The introduction can be improved by more background about cement and cement characterizations.

In equation number 1, what means the symbol m?

Moreover, the symbols of equation number 9 are not recognized.

The language needs more improvement.

Fig. 3b has not been cited within the text.

The compressive strength is one of the important indicators of durability and hydration of cement, why do the authors neglect this parameter?

The novelty and the aim of the study have to be clarified more.

Reviewer 2 Report

The manuscript entitled "Nondestructive Monitoring Hydration of Belite-Calcium Sulfoaluminate Cement by EIS Measurement" presents an interesting experimental study conducted on the evaluation/characterization of hydration kinetics of BCSA cement. However, the number of tested samples isn’t presented and many other issues must be addressed. The paper needs major revisions before it is processed further, some comments follow:

·       The abstract is written qualitatively. The majority of the qualitative statements should be modified for quantified result comparisons. Currently, the abstract only includes general/qualitative affirmations "The results prove that the resistivity of BCSA cement can reflect its microstructure".

·       The introduction section should be significantly improved. Please conduct a comprehensive and exhaustive study of the previous literature. Please clearly highlight the pros and cons of previous results and justify the need for the current research. Please discuss the highlights individually and assure a clear correspondence between the affirmations from the manuscript and those from the cited papers. Currently, the introduction section presents only general affirmation and base concepts about NDT method and concrete. Please consider the method and results presented in "DOI: 10.3390/ma14227018 and DOI: 10.1617/s11527-013-0139-9."

·       Table 1 - two types of iron oxides have been detected in these types of materials, therefore, please replace Fe2O3 with FexOy or provide the scientific proof to support your results. Moreover, which methods have been used to evaluate the properties presented in the table? Are these data obtained by the authors or they have been provided by the manufacturer (please introduce corresponding comments into the manuscript).

·       Isothermal calorimetry measurement. How many samples have been tested? In the case of mixed materials (multiple components), there is a possibility that the inhomogeneities from the materials result in different behavior. Please provide the standard deviation bar for each measurement.

·       Subsection 2.6. What soft was used to obtain the Nyquist and Bode diagrams, also the equivalent circuit? Please write this information.

·       The equivalent circuit model must be moved to the Results and discussion section.

·       Subsection 3.4 It is not enough to present only the Nyquist diagram, please also introduce the Bode plots. Also, introduce a table with the value of the parameters and discuss them. See an example: DOI: 10.3390/app10082753

·       The conclusions are too general "Linear relationship between the electrical resistivity and hydration heat, chemical shrinkage content or chemically bound water can be established, respectively." Please provide a brief quantitative analysis. Please provide the effect of w/c ratio on each property evaluated in this research.

Round 2

Reviewer 2 Report

The authors addressed most of my comments and the manuscript was improved accordingly. The paper can be processed further for publication.

Author Response

Thanks for your approval. If we find other minor faults, we will revise during proof process.